# Evaluation of a Medical Interview-Assistance System Using Artificial Intelligence for Resident Physicians Interviewing Simulated Patients: A Crossover, Randomized, Controlled Trial

**DOI:** 10.3390/ijerph20126176

**Published:** 2023-06-19

**Authors:** Akio Kanazawa, Kazutoshi Fujibayashi, Yu Watanabe, Seiko Kushiro, Naotake Yanagisawa, Yasuko Fukataki, Sakiko Kitamura, Wakako Hayashi, Masashi Nagao, Yuji Nishizaki, Takenori Inomata, Eri Arikawa-Hirasawa, Toshio Naito

**Affiliations:** 1Department of General Medicine, Faculty of Medicine, Juntendo University, Tokyo 113-8421, Japan; kfujiba@juntendo.ac.jp (K.F.); y.watanabe@juntendo.ac.jp (Y.W.); s-kushiro@juntendo.ac.jp (S.K.); ynishiza@juntendo.ac.jp (Y.N.); naito@juntendo.ac.jp (T.N.); 2Department of General Internal Medicine and Infectious Disease, Saitama Medical Center, Saitama Medical University, Saitama 350-8550, Japan; 3Medical Technology Innovation Center, Juntendo University, Tokyo 113-8421, Japan; n-yanagisawa@juntendo.ac.jp (N.Y.); nagao@juntendo.ac.jp (M.N.); 4Clinical Research and Trial Center, Juntendo University Hospital, Tokyo 113-8421, Japan; yafukataki@sapmed.ac.jp (Y.F.); s-kitamura@juntendo.ac.jp (S.K.); wakako.h1986@gmail.com (W.H.); 5Department of Ophthalmology, Juntendo University Graduate School of Medicine, Tokyo 113-8421, Japan; tinoma@juntendo.ac.jp; 6Department of Digital Medicine, Juntendo University Graduate School of Medicine, Tokyo 113-8421, Japan; 7Department of Hospital Administration, Juntendo University Graduate School of Medicine, Tokyo 113-8421, Japan; 8AI Incubation Farm, Juntendo University Graduate School of Medicine, Tokyo 113-8421, Japan; 9Department of Neurology, Faculty of Medicine, Juntendo University, Tokyo 113-8421, Japan; ehirasaw@juntendo.ac.jp

**Keywords:** decision-making, resident physician, artificial intelligence

## Abstract

Medical interviews are expected to undergo a major transformation through the use of artificial intelligence. However, artificial intelligence-based systems that support medical interviews are not yet widespread in Japan, and their usefulness is unclear. A randomized, controlled trial to determine the usefulness of a commercial medical interview support system using a question flow chart-type application based on a Bayesian model was conducted. Ten resident physicians were allocated to two groups with or without information from an artificial intelligence-based support system. The rate of correct diagnoses, amount of time to complete the interviews, and number of questions they asked were compared between the two groups. Two trials were conducted on different dates, with a total of 20 resident physicians participating. Data for 192 differential diagnoses were obtained. There was a significant difference in the rate of correct diagnosis between the two groups for two cases and for overall cases (0.561 vs. 0.393; *p* = 0.02). There was a significant difference in the time required between the two groups for overall cases (370 s (352–387) vs. 390 s (373–406), *p* = 0.04). Artificial intelligence-assisted medical interviews helped resident physicians make more accurate diagnoses and reduced consultation time. The widespread use of artificial intelligence systems in clinical settings could contribute to improving the quality of medical care.

## 1. Introduction

“Listen to your patient; he is telling you the diagnosis”, taught William Osler, the great physician of the late 19th century. Peterson reported that taking a medical history alone led to a final diagnosis in 76% of cases [1]. Many diagnoses can be predicted based solely on the patient’s medical history [2]. However, the medical interview, which involves listening to the patient’s medical history, is one of the most difficult clinical techniques to master. In addition, the limited consultation time for each patient is a global issue. Especially in the Japanese medical setting, physicians will generally examine more than a dozen outpatients during a morning or afternoon. It is important to collect sufficient information from patients efficiently in the shortest possible time. Physicians, no matter how skilled, cannot extract the entire history by themselves unaided, and patients who are usually inexperienced in illness cannot explain their history sufficiently by themselves without assistance [3]. It is difficult for inexperienced physicians to conduct appropriate medical interviews within a limited time. The interviews are thus collaborations between physicians and patients, and they often need further assistance.

Recently, it has been suggested that information technology may have the potential to improve clinical problem-solving, such as by reducing morbidity and improving clinical decision-making [4,5]. In the fields of radiology and pathology, clinical prediction modeling, medical artificial intelligence-based systems have been developed [6,7] and have also been expected to become established in clinical practice or complement healthcare providers during medical examinations, for example. A system using advanced machine learning algorithms that facilitates collaborations between physicians and patients, assisting in the extraction of necessary information from patients by adding appropriate questions, organizing that information chronologically, and presenting differential diagnoses may contribute to the creation of a better medical history.

Unfortunately, AI-based medical interviewing systems have not yet been widely used due to the complexity of their operational methods and the difficulty associated with their maintenance and version upgrades. However, a few medical interview systems have recently been launched in Japan. In this pilot study, a newly launched artificial intelligence-based medical interviewing system with a proprietary algorithm using a Bayesian model was evaluated.

We hypothesized that presenting results from a model constructed by an advanced machine learning algorithm based on appropriate data could improve the accuracy of differential diagnosis based on interviews and shorten the time required for those interviews. In our view, artificial intelligence-based decision support in the medical field will lead to standardization and improvement of the quality of medical care without having to rely on the experience of physicians. Experienced physicians are already more accurate with regard to diagnosis, with less variation compared to less-experienced ones such as resident physicians. Therefore, this study was designed based on the hypothesis that artificial intelligence provides particularly beneficial assistance to such less-experienced physicians, leading to our selection of resident physicians for the study as a sample.

To clarify the above hypothesis, medical interviews were conducted as a preclinical experiment with simulated patients and first-year resident physicians to evaluate the effectiveness of a practical artificial intelligence-based medical interview-assistance system using a proprietary algorithm based on a Bayesian model.

## 2. Materials and Methods

### 2.1. Study Design

This was a randomized, controlled, open-label, crossover, preclinical, pilot trial. The aim of this research was to evaluate the effectiveness, i.e., the improvement of the accuracy of diagnosis and the shortening of the time spent on the medical interview, of an artificial intelligence-based medical interview-assistance system. This study was conducted twice in 2019 at the Juntendo University Hospital General Medicine Department in Tokyo. The participants, first-year resident physicians from whom consent was obtained, were randomly assigned (1:1) to one of the study arms. After randomization, the participants interviewed the 10 simulated patients. The use of information from the artificial intelligence-based medical interview-assistance system, which is an intervention, was crossed over from the first half to the second half of each arm. The participants gave a diagnosis for each simulated patient, with and without information from the artificial intelligence-based medical interview-assistance system. As the main assessment measure, the participants gave three disease names as differential diagnoses. The primary endpoint was the accuracy rate of the differential diagnoses provided by the participant. The overall flow of this trial is shown in Figure 1.

### 2.2. Randomization, Blinding, and Crossover

Participants were randomly assigned to either interview with artificial intelligence-based system assistance for the medical history or the usual interview, at a 1:1 ratio, using a computer-generated random sequence. This sequence was created by a statistical analysis team member not involved in the statistical analyses of this study. Due to the study design, participants and simulated patients could not be blinded. The primary and secondary outcomes were crossed over and investigated in each group.

### 2.3. Study Interventions

In the study intervention group arm, an artificial intelligence-based medical interview-assistance system was used. This system alternates question selection and selects relevant diseases by a flowchart model of symptoms and diseases based on information from tens of thousands of medical reports (Ubie, Inc., Tokyo, Japan). The system was provided as a question flow chart-type application. The artificial intelligence-based medical interview system examined in the present study uses a proprietary algorithm based on a Bayesian model for data analysis. The system allowed patients to select categorical answers to natural language question statements, and the resulting categorical variables were analyzed as data. Following the flow chart, the program asks the user appropriate and relevant questions according to the answers from the patient as the user. A patient inputs information, such as his/her chief complaint, symptoms, and medical history prior to the interview for about 20 questions presented on a tablet. As the interview progressed, candidates for diseases with a certain degree of relevance were selected by the system. Then, questions highly relevant to the selected possible disease were selected. The artificial intelligence-based medical interview-assistance system selected questions according to the entered information, and when all questions were completed, the entered information was converted into a general medical chart format and displayed with a list of the 10 most relevant diseases. The resident physician in the intervention group viewed the printed output screen from the artificial intelligence-based medical interview support system during the medical interview. The obtained information was a document similar to a medical chart compiled by the artificial intelligence-based support system based on the medical inquiry about the disease and the medical history of the simulated patient entered by the researchers in advance. The resident physicians of the intervention group conducted their medical interviews with reference to these documents. The time allowed for medical interviews was limited to a maximum of 10 min. The resident physicians listed three differential diagnoses based on the information from the system and the information they obtained from the medical interview they conducted within the time limit. The items presented in the outcome section were compared between the intervention group and the control group.

In the control arm, the resident physicians conducted the interview without use of the artificial intelligence-based medical interview-assistance system, and they performed a usual medical interview of the simulated patient. The time allowed for each medical interview was limited to a maximum of 10 min. The resident physicians were asked to list three differential diagnoses based on only the information they obtained from the medical interview they conducted within the time limit. The experiment was conducted twice, with one set as follows: 10 resident physicians each interviewing 10 simulated patients and providing their diagnoses.

### 2.4. Outcome Measures

#### 2.4.1. Primary Outcome

The primary outcome was a significant difference in the correct answer rate for the differential diagnoses listed by the resident physician. A correct answer was defined as when any of the three differential diagnoses listed by the resident physician matched the disease played by the simulated patient. The correctness of the physicians’ differential diagnosis was judged by three collaborators—a physician, an orthopedist, and an ophthalmologist—all with more than 10 years of clinical experience. The collaborators judged whether the answers were correct and were blinded to the intervention state. If any of the collaborators’ judgments differed, the judgment was decided by majority decision. When controversial, the correctness of the answer was decided on in consultation with the corresponding author, who was blinded to the intervention state.

#### 2.4.2. Secondary Outcomes

Secondary outcomes were defined as follows:Significant difference in the time taken for each medical interview.Number of questions asked by the resident physician during a medical interview.

The duration and number of questions were measured by the simulated patients.

### 2.5. Participants’ Characteristics

In Japan, medical students are required to have six years of education. After graduation, medical students are allowed to engage in medical care simply by passing the national examination for medical practitioners. The medical abilities of first-year resident physicians therefore do not differ substantially from those of medical students. A two-year initial training period is required after obtaining a medical license, and the “super-rotation system” has been adopted, in which first-year resident physicians rotate through selected departments, with internal medicine, emergency medicine, and community medicine as compulsory. The purpose of this system is to train physicians to have the necessary skills to practice a wide range of medical care, especially primary care, regardless of their future specialty. Training in outpatient care has become more and more important in the super-rotation system in recent years.

Twenty first-year resident physicians were enrolled. Resident physicians were included in this study if they met at least one of the following inclusion criteria and had none of the exclusion criteria. The inclusion criteria were: first-year resident physicians who graduated from the university’s medical school in Japan; resident physicians who voluntarily applied to participate following the public announcement; and resident physicians who provided written, informed consent for participation in this study. The exclusion criteria were applicants considered ineligible by the researcher and participants who withdrew consent.

### 2.6. Simulated Patients

The simulated patients were selected according to the Post-Clinical Clerkship Objective Structured Clinical Examination (Post-CC OSCE) and were sent by My Informed Consent, Inc. (Tokyo, Japan). There were no conflicts of interest with My Informed Consent, Inc., and the research was not affected. Ten simulated patients each played patients with the following 10 diseases: case 1, infective endocarditis; case 2, polymyalgia rheumatica; case 3, hepatitis A; case 4, Guillain-Barré syndrome; case 5, multiple myeloma; case 6, acute myocarditis; case 7, pheochromocytoma; case 8, sarcoidosis; case 9, ischemic colitis; and case 10, renal infarction. The scenarios for the abovementioned diseases were created based on the questions of the Japanese National Medical Practitioners Qualifying Examination, and they were provided to the simulated patients. The simulated patients underwent a medical interview with participants based on the scenarios provided.

### 2.7. Ethics

This study was approved by the Ethics Committee of Juntendo University (approval number 2019043). Participants received written information about the trial, including its aim, expected advantages, and their role, and they were then asked to provide their written, informed consent. This study was retrospectively registered with University Hospital Medical Information Network (UMIN) under ID number UMIN000041435 on 1 September 2020. UMIN is a network member of the Japan Primary Registries Network (JPRN) as described in Primary registries, the WHO registry network. All procedures were performed in accordance with the relevant guidelines and regulations.

### 2.8. Statistical Methods and Sample Size Calculation

In this study, significance testing was conducted using a 2-sided test, and the significance level was set at 0.05. Values are summarized by means and 95% confidence intervals for continuous variables and counts for categorical variables. The number of correct diagnoses was compared between groups by Fisher’s exact test. The required time for a medical interview and the number of questions asked by the resident physicians were analyzed by the Mann–Whitney U test. All statistical analyses were performed using SAS software (version 9.4, SAS Institute Inc., Cary, NC, USA). The sample size was calculated as follows. Based on previous cases, the correct answer rate with assistance was estimated to be 90%. Since the correct diagnosis rate for trained physicians is generally 80%, the correct response rate for the control group without artificial intelligence assistance was set at 70%, with a difference of 20%. Even with the low correct answer rate of the resident physicians, the difference between the correct answer rate when they received artificial intelligence assistance and the control group was assumed to always be 20%. Under this assumption, the sample sizes were calculated for the control group at 30, 40, 50, 60, and 70% correct, respectively, and 93, 97, 93, 82, and 62 persons were needed for each group. Thus, the case needing the largest number of residents required 97, which means that 97 residents per group were needed. Since this was an exploratory study, and because of considerations regarding the feasibility of the study, 20 residents with and without artificial intelligence-based support, respectively, were set. The 20 resident physicians were to examine 5 simulated patients with AI support and 5 without AI support for a total of 200 answers. The 200 results were obtained by 20 resident physicians, 10 from each. However, the 10 simulated cases were independent, and the study was designed assuming that the individual results were independent. In addition, since this was an exploratory study, no adjustment for multiplicity was made.

## 3. Results

Two experiments were conducted on 19 October 2019 and 2 November 2019, respectively, and 10 resident physicians were enrolled for each experiment, for a total of 20 resident physicians. On 19 October 2019 and 2 November 2019, participating resident physicians were randomly assigned to two groups. In the medical interviews of simulated patients performed by the 20 resident physicians, the accuracy of the differential diagnosis, the time required for interviews, and the number of questions asked were evaluated. In the first test, one resident was unable to interview two cases with artificial intelligence-based assistance and five cases without artificial intelligence-based assistance. Therefore, in the first test, the results of 48 differential diagnoses with artificial intelligence-based assistance and 45 differential diagnoses without artificial intelligence-based assistance were obtained from interviews with 10 simulated patients (Figure 2). In the second test, one resident was unable to interview one case without artificial intelligence-based assistance. Therefore, in the second test, the results of 50 differential diagnoses with artificial intelligence-based assistance and 49 differential diagnoses without artificial intelligence-based assistance were obtained from interviews with 10 simulated patients (Figure 2). Finally, the results of the 192 differential diagnoses obtained were analyzed.

The correct diagnosis rates with and without information from the artificial intelligence-based support system for the two groups are shown in Table 1.

There was a significant difference in the rate of correct diagnosis between the two groups for two cases (case 7, 7/9 = 0.778 vs. 1/10 = 0.1, *p* = 0.01; and case 8, 7/10 = 0.7 vs. 1/10 = 0.1, *p* = 0.02) and for overall cases (55/98 = 0.561 vs. 37/94 = 0.393, *p* = 0.02). On the other hand, although no significant difference was seen, lower diagnostic accuracy rates were found for case 2 (5/10 = 0.5 vs. 6/9 = 0.667, *p* = 0.65), case 4 (5/10 = 0.5 vs. 8/9 = 0.889, *p* = 0.14), and case 10 (5/10 = 0.5 vs. 6/10 = 0.6, *p* = 1.00) in the group using the support system. In addition, the use of the support system seemed ineffective in case 5 (1/10 = 0.1 vs. 0/9, *p* = 1.00) and case 6 (2/7 = 0286. vs. 2/10 = 0.2, *p* = 1.00).

Comparisons of the time required and the number of questions with and without information from the artificial intelligence-based support system are shown in Table 2.

There was a significant difference in the time required between the two groups for overall cases (370 (352–387) seconds vs. 390 (373–406) seconds, *p* = 0.04). On the other hand, there were no significant differences in the number of questions between with the support system (40/37–43) and without the support system (41/38–43), *p* = 0.56).

## 4. Discussion

In the overall evaluation of this trial, resident physicians with artificial intelligence-based assistance derived a more accurate differential diagnosis through interviews with simulated patients, reducing the total interview time for all cases. On the other hand, individual cases showed some variability in results. The variability in results suggests that not all of the information provided by the support system was valid. The present results suggest that the use of the artificial intelligence-based interview support system can contribute to more efficient and accurate medical interviews by physicians but also suggest a need for caution regarding its use.

Previous studies of the effectiveness of artificial intelligence in assisting with medical interviews were limited, and few studies have been able to show the usefulness of diagnostic support systems using artificial intelligence [8,9]. On the other hand, in a few recent studies, computer-related diagnosis systems have achieved results in making problem lists or differential diagnoses [10,11,12,13]. Machine learning-related technologies are evolving at an amazing rate due to the increased amount of underlying information and the improvements in the algorithms used. Machine learning systems that improve themselves by accumulating clinical cases in both supervised and unsupervised learning are also being developed. Artificial intelligence-based diagnostic systems are rapidly improving, and the ability to analyze medical histories will thus continue to improve [14,15].

With regard to participants’ medical decision-making, the artificial intelligence-based medical interview support system appeared to be generally effective. On the other hand, individual cases showed some variability in results.

In more detail, the contribution of the artificial intelligence-based medical interview support system to the correct diagnosis was classified into four groups. First, in cases 7 and 8, the group using the artificial intelligence-based support system suggested a more accurate differential diagnosis. None of the following individual cases were shown to be statistically significant and should be considered with caution.

Second, in cases 1, 3, and 9, although there were no statistically significant differences, the group using the artificial intelligence-based support system may have suggested a more accurate differential diagnosis. Third, in cases 2, 5, 6, and 10, the use of the artificial intelligence-based support system may not have been related to more accurate differential diagnosis suggestions. Finally, in case 4, the group that did not use the artificial intelligence-based support system may have suggested a more accurate differential diagnosis. These results suggest the not all of the information provided by the artificial intelligence-based medical interview support system may have been helpful for diagnosis, or that the resident physicians were unable to use the information provided. Above all, it will be important to select appropriate cases to conduct clinical education using artificial intelligence-based systems, and experienced physicians should provide the learners with feedback on how to use artificial intelligence-based support.

The artificial intelligence-based medical interview support system needs a larger and more practical evaluation in the future.

In the present study, although there was no significant difference in the number of questions, the time required for history-taking tended to be shorter in each case, and there was a significant difference in total time; the results contrasted with a previously reported retrospective study [16]. Since the conditions of the present study differed from those in actual practice, it is difficult to compare the results with those of the previous report. The reduction in medical interview time observed in this study may not have a clinically significant impact. With artificial intelligence-based support, the reduction of time in each case was not very large, and the medical history information given to the resident physicians in advance by the artificial intelligence-based medical interview support system may have helped participants to complete the medical interview promptly.

It will also be necessary to evaluate the patient effort required to participate in medical interviews using artificial intelligence tools in the future. Whether a patient in poor health can properly communicate his or her information, including concerns and questions, through the AI interview system without stress should be evaluated.

## 5. Limitations

This study was only a preclinical experiment. Therefore, there are several important differences from actual medical interviews. First, simulated patients were used for this study. They may be able to convey their complaint or history of present illness better than real patients. Japan is aging rapidly, and the proportion of older patients is increasing among outpatients. A previous study showed that older patients tended to have difficulties with computer devices [17]. Real patients may not be able to answer the AI system’s questions well. In addition, the results of the artificial intelligence-based medical interview system were provided on printed paper to the participating resident physicians in this study. The use of a handout may affect test results since they differ from actual usage. Therefore, whether patients can use these devices in general clinical settings needs to be examined. Second, each clinical scenario was created based on the Japanese national examination for medical practitioners. Therefore, the medical interview scenarios used in the present study were characteristically typical of the diseases selected for the study. However, cases in the real world may be more complicated or involve multiple problems, and they also sometimes involve rare diseases. As a result, the medical interview scenarios used in the present study may have been simpler than a real-world medical history. In addition, there are cases in which the symptoms and the diagnosis actually diverge, as in the case of “indeterminate complaints”. In such cases, it may be difficult for artificial intelligence to make accurate predictions. Third, the present study involved small samples. In particular, the overall significant result for the effectiveness of the artificial intelligence-based support may have been influenced by a few cases in which it was highly effective. Therefore, the results may be biased. As such limitations have remained, data from real clinical situations need to be collected. Fourth, the AI system used in the present study was a commercial product; therefore, the specific algorithm, the dataset used, and the detailed evaluation are not disclosed. Finally, the results suggest that an artificial intelligence-based medical interview support system can influence the decision-making of resident physicians. In addition, due to the lack of post-usage interviews, the specific extent of the information’s utilization could not be determined. Artificial intelligence-based systems usually change their outputs depending on the data provided. The updated data change the results provided by the systems tested in the study. The present study’s results are thus not generalizable, as in the future different results may be obtained.

## 6. Conclusions

This pilot study showed the usefulness of an artificial intelligence-based support system in medical interviews. However, we need to be aware of the possibility of leading users to the wrong diagnosis in some cases. The technology of artificial intelligence-based support systems is constantly advancing. These devices will be increasingly used in real clinical situations [18]. We should thus proceed to further verify artificial intelligence-based support systems in various clinical settings.

## Figures and Tables

**Figure 1 ijerph-20-06176-f001:**
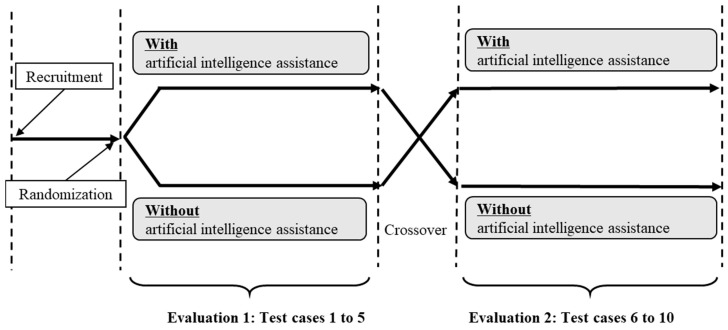
Study design; overall flow of the trial.

**Figure 2 ijerph-20-06176-f002:**
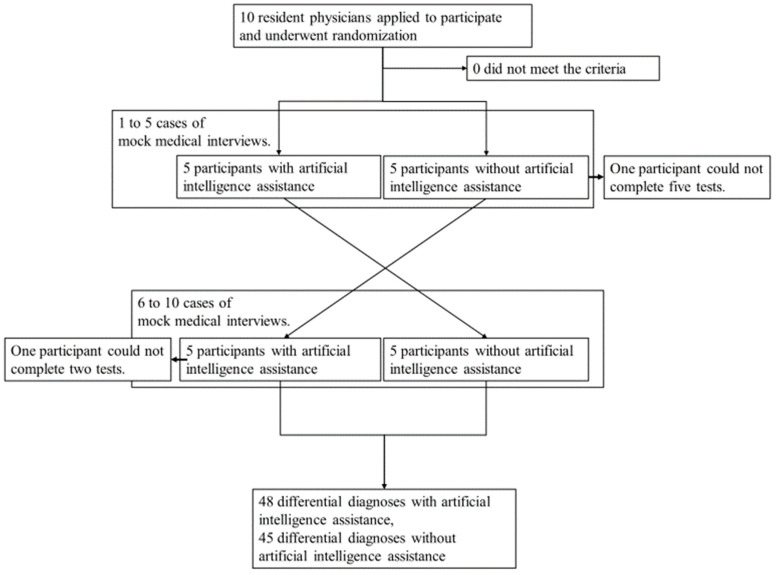
Enrollment and randomization, first and second test.

**Table 1 ijerph-20-06176-t001:** Comparison of correct diagnosis rates with and without information from the artificial intelligence-based support system.

With Information from the Support System Using Artificial Intelligence			Without Information from the Support System Using Artificial Intelligence		
	Correct Diagnosis (*n*)	Misdiagnosis (*n*)	Correct Diagnosis (*n*)	Misdiagnosis (*n*)	*p* Value
case 1	9	1	5	3	0.27
case 2	5	5	6	3	0.65
case 3	5	5	1	8	0.14
case 4	5	5	8	1	0.14
case 5	1	9	0	9	1.00
case 6	2	7	2	8	1.00
case 7	7	2	1	9	0.01 *
case 8	7	3	1	9	0.02 *
case 9	9	1	7	3	0.58
case 10	5	5	6	4	1.00
overall	55	43	37	57	0.02 *

Used Fisher’s exact test, *; *p* <0.05.

**Table 2 ijerph-20-06176-t002:** Comparison of doctors’ consultations with and without information from the artificial intelligence-based support system.

	With Information from the Support System Using Artificial Intelligence		Without Information from the Support System Using Artificial Intelligence		Time Required	Number of Questions
	Time Required (Seconds)	Number of Questions	Time Required (Seconds)	Number of Questions		
	Mean (95%CI)	Mean (95%CI)	Mean (95%CI)	Mean (95%CI)	*p* Value	*p* Value
case 1	350 (280–420)	40 (25–54)	400 (324–475)	47 (36–59)	0.25	0.26
case 2	352 (288–415)	37 (26–48)	385 (326–444)	40 (31–49)	0.27	0.57
case 3	393 (342–444)	45 (35–56)	403 (348–459)	43 (33–54)	0.78	0.78
case 4	385 (325–444)	32 (24–41)	389 (296–482)	31 (20–42)	0.38	0.78
case 5	388 (328–447)	45 (35–55)	378 (308–449)	39 (29–48)	0.97	0.45
case 6	367 (278–455)	38 (26–50)	394 (330–459)	41 (33–49)	0.60	0.66
case 7	373 (274–472)	44 (30–58)	366 (312–421)	42 (34–50)	0.84	0.87
case 8	387 (347–427)	42 (36–48)	429 (379–479)	48 (38–57)	0.15	0.48
case 9	363 (309–416)	38 (27–48)	376 (329–423)	34 (27–40)	0.55	0.77
case 10	340 (279–401)	38 (28–47)	378 (334–422)	41 (34–48)	0.46	0.63
overall	370 (352–387)	40 (37–43)	390 (373–406)	41 (38–43)	0.04 *	0.56

Used the Wilcoxon rank-sum test, *; *p* <0.05.

## Data Availability

Consistent with our institution’s ethics committee regulations, no further data are available.

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
