# Peer review of "Evaluation of a Medical Interview-Assistance System Using Artificial Intelligence for Resident Physicians Interviewing Simulated Patients: A Crossover, Randomized, Controlled Trial"

_ijerph, 2023, doi:10.3390/ijerph20126176_

Round 1
Reviewer 1 Report
This paper presents a randomised crossover evaluation of a commercially available "AI" system to assist clinical interviewing for differential diagnosis.
The research methodology used is appropriate and well described.
The results are presented and discussed appropriately.
The conclusions are appropriately drawn.
However, the paper fails to provide enough detail about the AI system used, and demonstrates a lack of familiarity with the history of AI research in the medical domain. Specific feedback section by section below.
Abstract
"Medical interviews are expected to undergo a major transformation through the use of artificial intelligence. However, research on the accuracy and effectiveness of artificial intelligence-based diagnosis systems is limited." - strange assertion to make. AI based diagnosis systems research goes back into the 1980s, and there's lots of papers excitedly claiming they're "very good" - the real catch is that very few of these research tools ever make it into practice, either because they don't fit workflows, they're overly specific, or they're incredibly cumbersome to maintain and update.
"A randomized, controlled trial to determine the usefulness of an artificial intelligence-based support system was conducted." - what AI based system? Should be stated.
Introduction
"To the best of our knowledge, no reports have evaluated the effectiveness of an artificial intelligence-based system for assisting the medical interview process." - do a literature review using the term "expert system" and you'll find many, in fact the concept of backward chaining expert systems was specifically developed for medical interviewing from the perspective of differential diagnosis i.e. you know they're sick but you ask a series of questions to narrow down what type of sickness. They will be mostly focused on a particular domain i.e. cardiovascular, or thyroid, or perhaps the most famous and first one of this type - bacterial infections (MYCIN), etc. but it was a huge area of research in the 80s in particular. The results were invariably quite good on paper, but they never got used in practice. Ultimately they were such a commercial failure that barely anyone continued researching through the 90s.
Trial design seems appropriate, but I note that the AI system is weakly described. More detail about how it was developed, the training dataset, how it's been previously evaluated etc. required.
It seems that the patients have to answer a bunch of questions from the AI system before even being given an opportunity to talk to the resident clinician. There is weakness here a) real patients may be very bad at answering the AI questions, b) real patients may be unable to answer the AI questions, c) patients time involved in getting a diagnosis is actually increased (time spent answering questionnaire plus time spent being interviewed by clinician). I'm noting this now because I expect to see these issues discussed later in the paper. I also expect to see the fundamental limitation discussed re: the simulated patient topics being potentially biased towards the kind of diagnoses the AI system is good at making, and the fact that real patients are usually much less clear cut and well defined compared to simulated patients.
Results
Table 2 is split over two pages
Discussion
The paragraph lines 346-357 is pretty speculative, and low value. Suggest removal.
"The reduction in medical interview time observed in this study may not make a large difference in actual medical situations [16]." - I think the term you're looking for here is "clinically insignificant". 20 seconds of clinician time saved on average in a "point in time" patient encounter.
"The medical history information given to the resident physicians in advance by the artificial intelligence-based medical interview support system may have helped to complete the medical interview promptly." - this is a key point - the AI system essentially just gave them a bit of medical history to start working with. In practical situations, the resident will pretty much always be given some kind of medical history to examine before the patient encounter on their chart, i.e. data collected by the triage nurse or data from previous encounters. Was this patient history/chart also provided in this simulated patient exercise? I don't think it was specified, but it would be normal for these kinds of simulated patients exercises.
I'd like to see a bit more discussion from the perspective of the potential patient, while acknowledging that you only had simulated patients. Paper focuses too much on the clinician perspective but forgets the patient's voice. What is it like to answer the questionnaire? Is it potentially frustrating if it asks you a bunch of seemingly irrelevant questions? Is it time consuming? Is it difficult to understand the questions? Does it provide appropriate options for the answer you select? Clinics will not implement this system if it annoys or upsets their patients, even if it does slightly improve diagnosis and clinician interview time.
Limitations
You do touch on most of the limitations I raised above, but I'd still suggest reviewing this section in light of the limitations I discussed above and see if you can't make it clearer.
Generally fine.
Author Response
Response to Reviewer 1
1.Abstract
"Medical interviews are expected to undergo a major transformation through the use of artificial intelligence. However, research on the accuracy and effectiveness of artificial intelligence-based diagnosis systems is limited." - strange assertion to make. AI based diagnosis systems research goes back into the 1980s, and there's lots of papers excitedly claiming they're "very good" - the real catch is that very few of these research tools ever make it into practice, either because they don't fit workflows, they're overly specific, or they're incredibly cumbersome to maintain and update.
"A randomized, controlled trial to determine the usefulness of an artificial intelligence-based support system was conducted." - what AI based system? Should be stated.
Response to your comment
Thank you for your comment. I have revised our abstract accordingly. As you point out, we are not sufficiently familiar with the history of the relationship between medicine and artificial intelligence. I have revised the manuscript as follows based on your suggestion.
“However, artificial intelligence-based systems that support medical interviews are not yet wide-spread in Japan, and their usefulness is unclear. A randomized, controlled trial to determine the usefulness of a commercial medical interview support system using a question flow chart-type application based on a Bayesian model was conducted.”
2.Introduction
"To the best of our knowledge, no reports have evaluated the effectiveness of an artificial intelligence-based system for assisting the medical interview process." - do a literature review using the term "expert system" and you'll find many, in fact the concept of backward chaining expert systems was specifically developed for medical interviewing from the perspective of differential diagnosis i.e. you know they're sick but you ask a series of questions to narrow down what type of sickness. They will be mostly focused on a particular domain i.e. cardiovascular, or thyroid, or perhaps the most famous and first one of this type - bacterial infections (MYCIN), etc. but it was a huge area of research in the 80s in particular. The results were invariably quite good on paper, but they never got used in practice. Ultimately they were such a commercial failure that barely anyone continued researching through the 90s.
Response to your comment
I have revised our introduction based on your comment, as below.
“A system using advanced machine learning algorithms that assists collaborations be-tween physicians and patients, assisting to extract necessary information from patients by adding appropriate questions, organizing that information chronologically, and present-ing differential diagnoses may contribute to the creation of a better medical history. Unfortunately, AI-based medical interviewing systems have not yet been widely used due to the complexity of their operational methods and difficulty associated with their maintenance and version upgrades. However, a few medical interview systems have recently been launched in Japan. In this pilot study, a newly launched artificial intelli-gence-based medical interviewing system with a proprietary algorithm using a Bayesi-an model was evaluated.
We hypothesized that presenting results from a model constructed by an advanced machine learning algorithm based on appropriate data could improve the accuracy of differential diagnosis based on interviews and shorten the time required for interviews. In our view, artificial intelligence-based decision support in the medical field will lead to standardization and improvement of the quality of medical care, without relying on the experience of physicians.”
3.Trial design seems appropriate, but I note that the AI system is weakly described. More detail about how it was developed, the training dataset, how it's been previously evaluated etc. required.
Response to your comment
Unfortunately, the AI system used in our study was a commercial product, and the company has not disclosed the critical information you indicated. We have added a note in the "Limitations" section that we were unable to obtain important information on the details of the AI system.
Fourth, the AI system used in the present study was a commercial product, and, therefore, the specific algorithm, the dataset used, and the detailed evaluation are not disclosed.
4.It seems that the patients have to answer a bunch of questions from the AI system before even being given an opportunity to talk to the resident clinician. There is weakness here a) real patients may be very bad at answering the AI questions, b) real patients may be unable to answer the AI questions, c) patients time involved in getting a diagnosis is actually increased (time spent answering questionnaire plus time spent being interviewed by clinician). I'm noting this now because I expect to see these issues discussed later in the paper. I also expect to see the fundamental limitation discussed re: the simulated patient topics being potentially biased towards the kind of diagnoses the AI system is good at making, and the fact that real patients are usually much less clear cut and well defined compared to simulated patients.
Response to your comment
As noted by the reviewer, we have added several sentences to the "Limitations" section.
“Real patients may not be able to answer the AI system’s questions well. In addition, the results of the artificial intelligence-based medical interview system were provided on printed paper to the participating resident physicians in this study.”
5.Results
Table 2 is split over two pages
Response to your comment
We have adjusted the table accordingly.
6.Discussion
The paragraph lines 346-357 is pretty speculative, and low value. Suggest removal.
Response to your comment
We have removed the relevant sentence from our manuscript as suggested.
- "The reduction in medical interview time observed in this study may not make a large difference in actual medical situations [16]." - I think the term you're looking for here is "clinically insignificant". 20 seconds of clinician time saved on average in a "point in time" patient encounter.
Response to your comment
We have made the following revisions to the manuscript, as suggested.
“The reduction in medical interview time observed in this study may not have a clinically significant impact [16]. With artificial intelligence-based support, the reduction of time in each case was not very large, the medical history information given to the resident physicians in advance by the artificial intelligence-based medical interview support system may have helped to complete the medical interview promptly.
8."The medical history information given to the resident physicians in advance by the artificial intelligence-based medical interview support system may have helped to complete the medical interview promptly." - this is a key point - the AI system essentially just gave them a bit of medical history to start working with. In practical situations, the resident will pretty much always be given some kind of medical history to examine before the patient encounter on their chart, i.e. data collected by the triage nurse or data from previous encounters. Was this patient history/chart also provided in this simulated patient exercise? I don't think it was specified, but it would be normal for these kinds of simulated patients exercises.
I'd like to see a bit more discussion from the perspective of the potential patient, while acknowledging that you only had simulated patients. Paper focuses too much on the clinician perspective but forgets the patient's voice. What is it like to answer the questionnaire? Is it potentially frustrating if it asks you a bunch of seemingly irrelevant questions? Is it time consuming? Is it difficult to understand the questions? Does it provide appropriate options for the answer you select? Clinics will not implement this system if it annoys or upsets their patients, even if it does slightly improve diagnosis and clinician interview time.
Response to your comment
Thank you for pointing out the inadequacies of our work. Following your suggestions, we have added the following recommendations for the future at the end of the Discussion.
It will also be necessary to evaluate the patient effort required to participate in medical interviews using artificial intelligence tools in the future. Whether a patient in poor health can properly communicate his or her information, including concerns and questions, through the AI interview system without stress should be evaluated.
9.Limitations
You do touch on most of the limitations I raised above, but I'd still suggest reviewing this section in light of the limitations I discussed above and see if you can't make it clearer.
Response to your comment
We have added several topics to the Limitations according to your suggestions.

Reviewer 2 Report
Review Report: Evaluation of a medical interview-assistance system using artificial intelligence for resident physicians interviewing simulated patients: A crossover, randomized, controlled trial
Minor comments:
The compartments used during 1-5 and then 6-10 cases can be designed using the networks.
This can help the readers to understand the hypothesis.
Major comments:
Introduction:
I think the authors can add a paragraph on the importance of AI in hospital settings and can add some useful relevant articles such as:
https://doi.org/10.1016/j.rinp.2022.105774
https://doi.org/10.1142/S0218348X22401223
https://doi.org/10.1007/s11063-022-10834-5
Methodology:
The research idea is really nice but it can be improved by taking into account some more cases. Currently the frequency is extremely low.
At this frequency, I am doubtful to agree with the findings.
Can you add some KNN tools to the algorithm for accuracy as the size is too small.
I think after addressing this important issue, the article can be considered for publication.
Thanks.
Review Report: Evaluation of a medical interview-assistance system using artificial intelligence for resident physicians interviewing simulated patients: A crossover, randomized, controlled trial
Minor comments:
The compartments used during 1-5 and then 6-10 cases can be designed using the networks.
This can help the readers to understand the hypothesis.
Major comments:
Introduction:
I think the authors can add a paragraph on the importance of AI in hospital settings and can add some useful relevant articles such as:
https://doi.org/10.1016/j.rinp.2022.105774
https://doi.org/10.1142/S0218348X22401223
https://doi.org/10.1007/s11063-022-10834-5
Methodology:
The research idea is really nice but it can be improved by taking into account some more cases. Currently the frequency is extremely low.
At this frequency, I am doubtful to agree with the findings.
Can you add some KNN tools to the algorithm for accuracy as the size is too small.
I think after addressing this important issue, the article can be considered for publication.
Thanks.
Author Response
Minor comments:
1.The compartments used during 1-5 and then 6-10 cases can be designed using the networks.
This can help the readers to understand the hypothesis.
Methodology:
At this frequency, I am doubtful to agree with the findings.
Can you add some KNN tools to the algorithm for accuracy as the size is too small.
I think after addressing this important issue, the article can be considered for publication.
Response to your comment
We appreciate your interesting and important comments regarding our manuscript. Unfortunately, the AI system we used in this study was developed by a company, so the detailed algorithm is not publicly available and not modifiable by us. In addition, the present study aimed to evaluate the impact of the developed AI system on medical interviews by comparing time taken, number of questions, and differential diagnoses, and, therefore, our goal was not to build a more accurate machine learning algorithm. I agree with your point that the small number of participants is a serious problem, and we would like to conduct future studies with more participants. Your interesting advice has provided important suggestions for our next research.
Thank you again.
Major comments:
2.Introduction:
I think the authors can add a paragraph on the importance of AI in hospital settings and can add some useful relevant articles such as:
https://doi.org/10.1016/j.rinp.2022.105774
https://doi.org/10.1142/S0218348X22401223
https://doi.org/10.1007/s11063-022-10834-5
Response to your comment
Thank you for introducing us to this important literature. I have added the following citations as references. Please see our revised manuscript.
“In the fields of radiology and pathology, clinical prediction modeling, medical artificial intelligence-based systems have been developed [6,7]”

Reviewer 3 Report
This is a generally well-written paper on a topical issue, namely AI-assistance during patient diagnosis. Specific comments follow:
Page 2, line 55: limited consultation time is not a problem limited to the Japanese medical setting, you could make a more general comment here.
Page 3, line 89: I’m not sure what a dynamic disease prediction model is – can you define please. Wouldn’t all disease models be dynamic?
Page 4, line 130: It is rather confusing talking about “disease candidates” here as you tend to think of this as a person (the patient?) not simply a set of possible diseases that may be indicated. I suggest rephrasing, “possible diseases”?
Page 5, line 189: “applicants considered ineligible by the researcher” – why would applicants be considered ineligible? Without details this sounds very arbitrary.
Page 9, line 291: You should not report your secondary outcomes before your primary outcome. The primary outcome is the overall number of cases diagnosed correctly. The secondary outcome is which individual cases were significantly improved. Technically you should use some kind of correction for multiple comparisons here, however as this is an exploratory study you should probably state that you have not done this.
Page 10, line 308: I think you have to be much more careful about concluding that some cases or some information provided by the AI was “useless” for the physician. Ideally a follow up interview could confirm which pieces of information physicians found useful or not, but you haven’t collected this, so you should just say that there was some variability in results, suggesting the not all of the information provided by the AI may have assisted the diagnosis.
Page 10, line 323: “become smarter every day” – this seems like hyperbole. Better to say are rapidly improving.
Page 10, last paragraph: As I said above without further information you should be more cautious about drawing conclusion about which cases were useful or not useful – simply saying that variability was seen is really all you can say.
Page 11, line 338: The point about case 4 apparently making diagnosis worse is interesting, but again this is only one case, and so you need to be cautious about drawing too many conclusions from one case.
Page 12, line 420: “access to information being severely restricted” – this is a very odd phrase, which suggests that the authors are not happy with the decisions of their ethics committee? Better to say something like “consistent with our institution’s ethics committee regulations, no further data is available”.
END
Just a few rephrasings are needed, as described in comments above
Author Response
Response to Reviewer 3
- Page 2, line 55: limited consultation time is not a problem limited to the Japanese medical setting, you could make a more general comment here.
Response to your comment
Thank you for your comment. In response to your feedback, we have highlighted the global nature of the issue.
“However, the medical interview, listening to the patient’s medical history, is one of the most difficult clinical techniques to master. In addition, the limited consultation time for each patient is a global issue.”
- Page 3, line 89: I’m not sure
what a dynamic disease prediction model is – can you define please. Wouldn’t all disease models be dynamic?
Response to your comment
We apologize for the ambiguous and confusing description. We have revised our manuscript as follows.
“a practical artificial intelligence-based medical interview-assistance system using a proprietary algorithm based on a Bayesian model.”
- Page 4, line 130: It is rather confusing talking about “disease candidates” here as you tend to think of this as a person (the patient?) not simply a set of possible diseases that may be indicated. I suggest rephrasing, “possible diseases”?
Response to your comment
Taking your feedback into account, we have made the following modifications to enhance clarity.
“Then, questions highly relevant to the selected possible disease were selected.”
- Page 5, line 189: “applicants considered ineligible by the researcher” – why would applicants be considered ineligible? Without details this sounds very arbitrary.
Response to your comment
We deemed those unable to participate in the examination schedule as ineligible. In fact, no participants were disqualified in this study.
- Page 9, line 291: You should not report your secondary outcomes before your primary outcome. The primary outcome is the overall number of cases diagnosed correctly. The secondary outcome is which individual cases were significantly improved.
Response to your comment
It seems there has been a misunderstanding. We have indeed listed the Primary outcome before the Secondary outcome. If there has been any misinterpretation in our response, please let us know.
- Technically you should use some kind of correction for multiple comparisons here, however as this is an exploratory study you should probably state that you have not done this.
Response to your comment
Since this study was an exploratory study, no adjustment for multiplicity was performed. We have revised our manuscript to describe this as follows.
“In addition, since this was an exploratory study, no adjustment for multiplicity was made.”
- Page 10, line 308: I think you have to be much more careful about concluding that some cases or some information provided by the AI was “useless” for the physician. Ideally a follow up interview could confirm which pieces of information physicians found useful or not, but you haven’t collected this, so you should just say that there was some variability in results, suggesting the not all of the information provided by the AI may have assisted the diagnosis.
Response to your comment
Upon further consideration of your feedback, we have added to the Discussion and Limitations sections the following to address the points you raised:
Discussion section
“In the overall evaluation of this trial, resident physicians with artificial intelligence-based assistance derived a more accurate differential diagnosis through interviews with simulated patients, reducing the total interview time for all cases. On the other hand, individual cases showed some variability in results. The variability in results suggests that not all of the information provided by the support system was valid. The present results suggest…”
Limitations section
“In addition, due to the lack of post-usage interviews, the specific extent of the infor-mation’s utilization cannot be determined.”
- Page 10, line 323: “become smarter every day” – this seems like hyperbole. Better to say are rapidly improving.
Response to your comment
I appreciate your feedback regarding the wording used. We have revised our manuscript as follows.
“Machine learning systems that improve themselves by accumulating clinical cases in both supervised or unsupervised learning are also being developed. Artificial intelligence based diagnostic systems are rapidly improving, and the ability to analyze medical histories will thus likewise continue to improve [14,15].”
- Page 10, last paragraph: As I said above without further information you should be more cautious about drawing conclusion about which cases were useful or not useful – simply saying that variability was seen is really all you can say.
- Page 11, line 338: The point about case 4 apparently making diagnosis worse is interesting, but again this is only one case, and so you need to be cautious about drawing too many conclusions from one case.
Response to your comment
We agree with your call for caution. Our study was strictly a pilot study. We intend to conduct further validation and exploration in the future.
- Page 12, line 420: “access to information being severely restricted” – this is a very odd phrase, which suggests that the authors are not happy with the decisions of their ethics committee? Better to say something like “consistent with our institution’s ethics committee regulations, no further data is available”.
Response to your comment
Following your guidance, we have revised our manuscript as follows.
Data Availability Statement: “Consistent with our institution’s ethics committee regulations, no further data are available.”
Round 2
Reviewer 2 Report
The authors have revised the article nicely and can be accepted in present form.
The authors have revised the article nicely and can be accepted in present form.